# AN ANALYSIS OF FEATURE REGULARIZATION FOR LOW-SHOT LEARNING

**Zhuoyuan Chen, Xiao Liu & Wei Xu**
Baidu Research
Sunnyvale, CA 94089, USA
{chenzhuoyuan,liuxiao,wei.xu}@baidu.com

**Han Zhao**
Department of Computer Science
Carnegie Mellon University
Pittsburgh, PA 15213
han.zhao@cs.cmu.edu

## ABSTRACT

Low-shot visual learning, the ability to recognize novel object categories from very few, or even one example, is a hallmark of human visual intelligence. Though successful on many tasks, deep learning approaches tends to be notoriously data-hungry. Recently, feature penalty regularization has been proved effective on capturing new concepts. In this work, we provide both empirical evidence and theoretical analysis on how and why these methods work. We also propose a better design of cost function with improved performance. Close scrutiny reveals the centering effect of feature representation, as well as the intrinsic connection with batch normalization. Extensive experiments on synthetic datasets, the one-shot learning benchmark "Omniglot", and large-scale ImageNet validate our analysis.

## 1 INTRODUCTION

The current success of deep learning hinges on the ability to apply gradient-based optimization to high-capacity models. It has achieved impressive results on many large-scale supervised tasks such as image classification Krizhevsky et al. (2012); He et al. (2016) and speech recognition Yu & Deng (2012). Notably, these models are extensively hungry for data.

In contrast, human beings have strong ability to learn novel concepts efficiently from very few or even one example. As pointed out in Lake et al. (2016), human learning is distinguished by its richness and efficiency. To test whether machines can approach this goal, Lake *et al.* propose the invaluable "Omniglot" hand-written character classification benchmark Lake et al. (2011), where each training class has very few examples and the ability to fast-learn is evaluated on never-seen classes with only one example.

There has been previous work on attaining rapid learning from sparse data, denoted as meta-learning or learning-to-learn Thrun. (1998); Baxter. (1998). Although used in numerous senses, the term generally refers to exploiting meta-knowledge within a single learning system across tasks or algorithms. In theory, a meta-learning is able to identify the right "inductive bias shifts" from previous experiences given enough data and many tasks Baxter. (1998). However, even if a well-designed convolutional neural network is a good "inductive bias shift" for a visual recognition task, it is still elusive to find the optimal parameter from a small training set without any prior knowledge.

To alleviate this issue, low-shot learning methods have been proposed to transfer knowledge from various *priors* to avoid over-fitting, such as Bart & Ullman (2005). Recently, Hariharan & Girshick. (2016) propose a novel prior of *gradient penalty*, which works pretty well experimentally. Although an intuitive explanation is provided in Hariharan & Girshick. (2016) that a good solution of the network parameters should be stable with small gradients, it is mysterious why adding such a regularization magically improves the low-shot task by a large margin. Mathematical derivation shows that gradient penalty is closely related to *regularizing the feature representation*.

In this paper, we give more analysis, both empirically and theoretically, on why adding a ***gradient regularization***, or ***feature penalty*** performs so well. Moreover, we carefully carry out two case studies: (1) the simplest non-linear-separable XOR classification, and (2) a two-layer linear network for regression. The study does give insight on how the penalty *centers* feature representations and

make the learning task easier. Furthermore, we also theoretically show that adding another final-layer weight penalty is *necessary* to achieve better performance. To be added, close scrutiny reveals its *inherent connection* with *batch normalization* Ioffe & Szegedy (2015). From a Bayesian point of view, feature penalty essentially introduces a Gaussian prior which softly normalizes the feature representation and eases the following learning task.

## 1.1 RELATED WORK

There is a huge body of literature on one-shot learning and it is beyond the scope of this paper to review the entire literature. We only discuss papers that introduce prior knowledge to adjust the neural network learning process, as that is the main focus of this work.

Prior knowledge, or "inductive bias" Thrun. (1998), plays an important role in one-shot or low-shot learning. To reduce over-fitting problems, we need to regularize the learning process. Common techniques include weight regularization Bishop (1995). Recently in Hariharan & Girshick. (2016), a gradient penalty is introduced, which works well experimentally in low-shot scenarios.

Various forms of feature regularization have been proposed to improve generalization: Dropout Srivastava et al. (2014) is effective to reduce over-fitting, but has been eschewed by recent architectures such as batch-normalization Ioffe & Szegedy (2015) and ResNets He et al. (2016). Other forms have also been proposed to improve transfer learning performance, such as minimizing the correlation of features Cogswell et al. (2016) and the multiverse loss Littwin & Wolf (2015) .

Our work is also closely related to metric learning and nearest neighbor methods, in which representations from previous experience are applied in cross-domain settings Fink (2005); Koch et al. (2015); Goldberger et al. (2005); Chopra et al. (2005). The insight lies in that a well-trained representational model have strong ability to generalize well on new tasks. In a recent work Santoro et al. (2016), DeepMind proposed a Memory Augmented Neural Network (MANN) to leverage the Neural-Turing-Machine for one-shot tasks. However, in Vinyals et al. (2016), it is found that a good initialization such as VGG-Net largely improves the one-shot performance. In our opinion, a good feature representations still play a central role in low-shot tasks.

## 1.2 CONTRIBUTIONS

The main contributions of our comprehensive analysis are three-fold:

1. We carefully carry out two case studies on the influence of feature regularization on shallow neural networks. We observe how the regularization centers features and eases the learning problem. Moreover, we propose a better design to avoid degenerate solutions.
2. From Bayesian point of view, close scrutiny reveals internal connections between feature regularization and batch normalization.
3. Extensive experiments on synthetic, the "Omniglot" one-shot and the large-scale ImageNet datasets validate our analysis.

## 2 AN ANALYSIS OF FEATURE REGULARIZATION

We briefly introduce the notations in our work: we denote uppercase $A$, bold $\mathbf{a}$ and lowercase $a$ for matrices, vectors and scalars respectively. For a vector $\mathbf{a}_i$, we denotes $\mathbf{a}_{i,j}$ as its $j$-th element. $||.||_F$ stands for the Frobenius norm of a matrix. Given $N$ examples $\{(\mathbf{x}^i, y^i)|i = 1, ..., N\}$, we define $\mathbb{E}\{.\}$ as an expectation taken with respect to the empirical distribution generated by the training set.

Following Hariharan & Girshick. (2016), we aim to learn a neural network model to extract the feature representations $\phi(\mathbf{x}^i)$ and make predictions $\hat{y}_i = W\phi(\mathbf{x}^i)$ with $W = [\mathbf{w}_1, ..., \mathbf{w}_{|C|}]$. This setting includes both classification and regression problems, with $|C|$ as the number of classes or target dimension, respectively. The problem can be generally formulated as:

$$W^*, \phi^* = \arg\min_{W,\phi} \mathbb{E}\{l(W, \phi(\mathbf{x}^i), y^i)\} \tag{1}$$

where $l(.)$ can be any reasonable cost function. In this paper, we focus on cross-entropy and $L_2$ loss due to their convexity and universality.

In Hariharan & Girshick. (2016), it is suggested that adding a squared gradient magnitude loss (SGM) on every sample can regularize the learning process.

$$W^*, \phi^* = \arg \min_{W,\phi} \mathbb{E}\{l(W, \phi(\mathbf{x}^i), y^i) + \lambda ||\nabla_W l(W, \phi(\mathbf{x}^i, y^i))||^2\} \tag{2}$$

The insight is that for a good solution, the parameter gradient should be small at convergence. However, we know that the convergence of a neural network optimization is a dynamic equilibrium. In other words, at a stationary point, we should have $\mathbb{E}\{\nabla_W l(W, \phi(\mathbf{x}))\} \to 0$. Intuitively when close to convergence, about half of the data-cases recommend to update a parameter to move positive, while the other half recommend to move negative. It is not very clear why *small gradients on every sample* $\mathbb{E}\{||\nabla_W l(W, \phi(\mathbf{x}))||^2\}$ produces good generalization experimentally.

Mathematical derivation shows that the optimization problem with gradient penalty is equivalent with adding a weighted $L_2$ regularizer $\phi(\mathbf{x}^i)$:

$$\arg \min_{W,\phi} \mathbb{E}\{l(W, \phi(\mathbf{x}^i), y^i) + \lambda \alpha^i ||\phi(\mathbf{x}^i)||^2\} \tag{3}$$

where the example-dependent $\alpha^i$ measures the deviation between the prediction $\hat{y}^i$ and the target $y^i$. In a regression problem, we have $\alpha^i = r^2 = ||\hat{y}^i - y^i||^2$, with the residual $r = \hat{y}^i - y^i$; in a classification problem, we have $\alpha^i = \sum_k (p_k^i - I(y^i = k))^2$. Intuitively, the misclassified high-norm examples might be outliers, and in a low-shot learning scenario, such outliers can pull the learned weight vectors far away from the right solution. In Hariharan & Girshick. (2016), the authors compare dropping $\alpha^i$ and directly penalizing $||\phi(x^i)||^2$, which performs almost equally well.

In our work, we argue that a better and more reasonable design should be:

$$\arg \min_{W,\phi} \mathbb{E}\{l(W, \phi(\mathbf{x}^i), y^i) + \lambda_1 \alpha^i ||\phi(\mathbf{x}^i)||^2\} + \lambda_2 ||W||_F^2 \tag{4}$$

where we add another weight regularizer $||W||_F^2$, which is necessary to avoid degenerate solutions. We will give further explanation in our second case study. In following analysis, we denote the cost in Eqn(4) with example-dependent $\alpha^i$ as **weighted $L_2$ feature penalty**, and the example-independent (setting $\alpha^i \equiv 1$) as **uniform $L_2$ feature penalty**.

We carry out two case studies: (1) an XOR classification and (2) a regression problem, both empirically and theoretically to analyze how the uniform and weighted $L_2$ feature penalty regularize the neural network. In our paper, we will focus on the **uniform** feature regularization and will also cover the **weighted** scenario as well.

## 2.1 Case Study 1: an Empirical Analysis of XOR Classification

First, we study the simplest linear-non-separable problem– exclusive-or (XOR). Suppose that we have four two-dimensional input points $\mathbf{x} = [x_1, x_2]^T \in R^2$: $\{\mathbf{x}^1 = [1,1]^T, \mathbf{x}^2 = [0,0]^T\} \in C_+$ belongs to the positive class, while $\{\mathbf{x}^3 = [1,0]^T, \mathbf{x}^4 = [0,1]^T\} \in C_-$ the negative.

As shown in Figure 1, we use a three-layer neural network to address the problem (left figure): $\mathbf{h_1} = \mathbf{x} + \mathbf{b}$ is a translation with $\mathbf{b} = [b_1, b_2]^T \in R^2$ as the offset; $h_2 = h_{1,1} * h_{1,2}$ is a non-linear layer, multiplying the first and second dimension of $\mathbf{h}_1$ and producing a scalar; $y$ is a linear classifier on $h_2$ parametrized by $w$. The original classification problem can be formulated as:

$$\arg \min_{w_1, \mathbf{b}, w} \sum_{i=1}^{4} \log(1 + \exp(-y^i * w * h_2^i))$$

Suppose that we start from an initialization $\mathbf{b} = [0,0]^T$, all three samples $\{\mathbf{x}^2, \mathbf{x}^3, \mathbf{x}^4\}$ from different classes will produce the same representation $h_2 = 0$, which is not separable at all. It takes efforts to tune the learning rate to back-propagate $w, \mathbf{b}$ updates from the target $y$.

However, if we introduce the **uniform $L_2$** feature regularization as:

$$\arg \min_{w_1, \mathbf{b}, w} \sum_{i=1}^{4} \log(1 + \exp(-y^i w h_2^i)) + \frac{\lambda_1}{2} ||h_2^i||^2$$

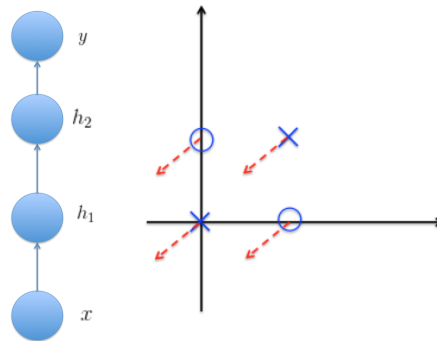

Figure 1: Case 1: an empirical study of the XOR classification task. **The left figure:** the network structure we use: $h_1$ is a linear transformation, $h_2$ is a non-linear transform of $h_1$ and $y$ is the prediction; **The right column:** The linear transformation maps $\mathbf{x}$ to $\mathbf{h_1}$. As shown in the red arrow, an $L_2$ norm penalty on $h_2$ centers the feature of $h_1$ and make the points from different sets separable. 'X's refer to positive examples, and 'O's are negative ones.

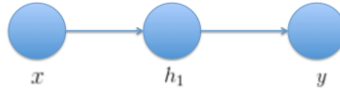

Figure 2: Case 2: a comprehensive study of a two-layer linear neural network for regression task. We minimize the $L_2$ distance between the prediction $\hat{y} = W_2(W_1\mathbf{x} + \mathbf{b}_1) + \mathbf{b}_2$ and $y$. The latent representation $\mathbf{h} = W_1\mathbf{x} + b_1$ is a linear mapping.

Then, we have:

$$\frac{\lambda_1}{2}\frac{\partial ||h_2||^2}{\partial \mathbf{b}} = \lambda_1 \begin{pmatrix} \mathbb{E}\{h_2(x_2 + b_2)\} \\ \mathbb{E}\{h_1(x_1 + b_1)\} \end{pmatrix} \tag{5}$$

the gradient descent pulls Eqn(5) towards zero, i.e., pulling $\mathbf{b}$ towards $b_1 = -\mathbb{E}\{x_1\} = -0.5$ and $b_2 = -\mathbb{E}\{x_2\} = -0.5$.

As shown on the right of Figure 1, the gradient of feature regularization pulls $\mathbf{h_1}$ along the direction of red arrows. Then, we have $h_2 > 0$ for positive examples and $h_2 < 0$ for negative ones, which means $h_2$ is linearly-separable. In summary, we can observe that:
**Empirically, the feature regularization *centers* the representation $h_2 = \phi(\mathbf{x})$ and makes the following classification more learnable**.
For the weighted case, the offset $\mathbf{b}$ have similar effects. It can be derived that when converged the feature representation will satisfy $\mathbb{E}\{\mathbf{h_1}\} = \mathbf{0}$ and $\mathbb{E}\{h_2\}=0$.

## 2.2 CASE STUDY 2: A COMPREHENSIVE ANALYSIS ON A REGRESSION PROBLEM

Next, we analyze a two-layer linear neural network as shown in Figure 2. Denoting the input as $X = [\mathbf{x}_1, \mathbf{x}_2, ...]$ and the target as $Y = [\mathbf{y}_1, \mathbf{y}_2, ...]$. The regression loss can be formulated as:

$$\mathbf{E}\{||y - W_2W_1x||^2\}$$

where the latent feature is $\mathbf{h} = \phi(\mathbf{x}) = W_1\mathbf{x}$. The optimization of $\{W_1, W_2\}$ in this multi-layer linear neural network is not trivial, since it satisfies following properties:

1. The regression loss is non-convex and non-concave. It is convex on $W_1$ (or $W_2$) when the other parameter $W_2$ (or $W_1$) is fixed, but not convex on both simultaneously;

2. Every local minimum is a global minimum;

3. Every critical point that is not a global minimum is a saddle point;

4. If $W_2 * W_1$ is full-rank, the Hessian at any saddle point has at least one negative eigenvalue.

We refer interested readers to Baldi & Hornik (1989); Kawaguchi (2016) for detailed analysis.

In case of the uniform $L_2$ feature penalty, the problem becomes:

$$E(W_1, W_2) = \mathbb{E}\{\frac{1}{2}||\mathbf{y} - W_2 W_1 \mathbf{x}||^2 + \frac{\lambda_1}{2}||W_1\mathbf{x}||^2\} + \frac{\lambda_2}{2}||W_2||_F^2 \qquad (6)$$

At the global minimum $\{W_1^*, W_2^*\}$, we should have:

$$\frac{\partial E}{\partial W_1}|_{W_1^*} = W_2^T \Sigma_{xy} - W_2^T W_2 W_1 \Sigma_{xx} + \lambda_1 W_1 \Sigma_{xx} = 0 \qquad (7)$$

$$\frac{\partial E}{\partial W_2}|_{W_2^*} = \Sigma_{xy} W_1^T - W_2 W_1 \Sigma_{xx} W_1^T + \lambda_2 W_2 = 0 \qquad (8)$$

where we define the variance and covariance matrix as $\Sigma_{xx} = \mathbb{E}\{\mathbf{x}\mathbf{x}^T\}$, $\Sigma_{xy} = \mathbb{E}\{\mathbf{y}\mathbf{x}^T\}$. Carrying out $Eqn(7) * W_1^T - W_2^T * Eqn(8) = 0$ reveals a very interesting conclusion:

$$\lambda_1 \mathbb{E}\{||W_1\mathbf{x}||^2\} = \lambda_2 ||W_2||_F^2 \qquad (9)$$

This reads as ***the expected $L_2$ feature penalty should be equal to final-layer weight regularizer when converged***. Or equivalently, when close to convergence, the $L_2$ feature penalty reduces overfitting by implicitly penalizing the corresponding weight matrix $W$. A more generalized form is:

**Lemma 1** *For a cost function of form in Eqn (4) with uniform $L_2$ feature regularization:*

$$\arg\min_{W,\phi} \mathbb{E}\{l(W, \phi(\mathbf{x}^i), y^i) + \lambda_1||\phi(x_i))||^2\} + \lambda_2||W||_F^2$$

*we have:*

$$\lambda_1 \mathbb{E}\{||\phi(\mathbf{x})||^2\} = \lambda_2 ||W||_F^2 \qquad (10)$$

The $\phi(.)$ can take a quite general form of a convolutional neural network with many common non-linear operations such as the ReLU, max-pooling and so on. One can follow the derivation of Eqn(9) to easily derive **Lemma 1**.

Lemma 1 also reveals the importance of adding the weight penalty $||W||_F^2$ in Eqn(4). If we only include the the feature penalty and drop the weight penalty ($\lambda_2 = 0$ in our case), then a scaling as $\phi(.) = \gamma\phi(.)$ and $W = \frac{1}{\gamma}W$ with $\gamma < 1$ will always decrease the energy and the solution will become very ill-conditioned with $\gamma \to 0$.

### 2.2.1 $L_2$ FEATURE REGULARIZATION MAKES OPTIMIZATION EASIER

Moreover, we analyze numerically how the $L_2$ feature penalty influences the optimization process in our regression problem.

We study a special case $\{\mathbf{x} \in \mathbb{R}^d, y \in \mathbb{R}\}$ with $W_1 \in \mathbb{R}^{1 \times m}$, $W_2 \in \mathbb{R}$ and include offsets $b_1 \in \mathbb{R}$ and $b_2 \in \mathbb{R}$ to make the problem more general. Then, the latent representation becomes $h = W_1\mathbf{x} + \mathbf{b}_1$ and the prediction is $\hat{y} = W_2 h + b_2$. The cost function of Eqn(4) becomes:

$$E(W_1, b_1, W_2, b_2) = \frac{1}{2}\mathbb{E}\{(W_2 h + b_2 - y)^2 + \frac{\lambda_1}{2}r_\dagger^2 h^2\} + \frac{\lambda_2}{2}W_2^2 \qquad (11)$$

We define the prediction residual $r$ and $r_\dagger$ as $\hat{y} - y$ for better readability, and substitute $\alpha^i = (r_\dagger^i)^2$. Numerically, we apply a **two-step** process: in the first step, we calculate the sample-dependent $r_\dagger^i$ in the feed-forward process to obtain our $L_2$ feature regularization weights $\alpha^i$ for each $(\mathbf{x}^i, y^i)$; in the second step, we treat $r_\dagger$ as a constant in the optimize. The gradient and Hessian matrix of Eqn(11) can be derived as:

$$\frac{\partial E}{\partial W_2} = \mathbb{E}\{rh^T\} + \lambda_2 W_2 \qquad \frac{\partial E}{\partial b_2} = \mathbb{E}\{r\} \qquad \frac{\partial E}{\partial h^i} = W_2 r^i + \lambda_1(r_\dagger^i)^2 h^i$$

$$\frac{\partial E}{\partial W_1} = W_2 \mathbb{E}\{r\mathbf{x}^T\} + \lambda_1 \mathbb{E}\{r_\dagger^2 h\mathbf{x}^T\} \qquad \frac{\partial E}{\partial b_1} = W_2 \mathbb{E}\{r\} + \lambda_1 \mathbb{E}\{r_\dagger^2 h\}$$

and

$$\frac{\partial^2 E}{\partial W_2^2} = \mathbb{E}\{hh^T\} + \lambda_2 \qquad\qquad \frac{\partial^2 E}{\partial b_2^2} = 1 \qquad\qquad \frac{\partial^2 E}{\partial (h^i)^2} = W_2^2 + \lambda_1 (r_\dagger^i)^2$$

$$\frac{\partial^2 E}{\partial W_1^2} = (W_2^2 + \lambda_1 \mathbb{E}\{r_\dagger^2\})\mathbb{E}\{\mathbf{x}\mathbf{x}^T\} \qquad \frac{\partial^2 E}{\partial b_1^2} = W_2^2 + \lambda_1 \mathbb{E}\{r_\dagger^2\}$$

Suppose that we apply a second-order optimization algorithm for the network parameter $\theta \in \{w_1, w_2, b_1, b_2\}$, the updates should be $\triangle \theta = -\eta * (\partial^2 E / \partial \theta^2)^{-1} (\partial E / \partial \theta)$ with $\eta > 0$ as the step-size. If we unluckily start from a bad initialization point $W_2 \to 0$, the updates of $h_i$, $W_1$ and $b_1$ are of the form:

$$\triangle h^i = -\eta(W_2^2 + \lambda_1(r_\dagger^i)^2\})^{-1}(W_2 r^i + \lambda_1 (r_\dagger^i)^2 h^i)$$
$$\triangle \mathbf{w}_1 = -\eta[(W_2^2 + \lambda_1 \mathbb{E}\{r_\dagger^2\})\mathbb{E}\{\mathbf{x}\mathbf{x}^T\}]^{-1}$$
$$(W_2 \mathbb{E}\{r\} + \lambda_1 \mathbb{E}\{r_\dagger^2 h\})\mathbb{E}\{\mathbf{x}\}$$
$$\triangle b_1 = -\eta(W_2^2 + \lambda_1 \mathbb{E}\{r_\dagger^2\})^{-1}(W_2 \mathbb{E}\{r\} + \lambda_1 \mathbb{E}\{r_\dagger^2 h\})$$

The updates will become very ill-conditioned without the regularization term ($\lambda_1 = 0$), since spectrum of the Hessian matrix is two-orders of infinitesimal $O(W_2^2)$ and the gradient is of one-order $O(W_2)$. In comparison, with a reasonable choice of $\lambda_1 > 0$, the computation can be **largely stabilized** when $\mathbb{E}\{r_\dagger^2\} \neq 0$.

When the algorithm finally converges to a local minimum $\{W_1^*, b_1^*, W_2^*, b_2^*\}$, the expectation of parameter and latent feature should have gradient close to $\mathbf{0}$:

$$\frac{\partial E}{\partial W_2} = \mathbb{E}\{rh^T\} + \lambda_2 W_2 \to 0 \qquad\qquad \frac{\partial E}{\partial b_2} = \mathbb{E}\{r\} \to 0$$

Substituting this in the analysis of $b_1$, we have:

$$\frac{\partial E}{\partial b_1} = W_2 \mathbb{E}\{r\} + \lambda_1 \mathbb{E}\{\alpha h\} \implies \mathbb{E}\{\alpha h\} \to 0 \tag{12}$$

In other words, the feature penalty **centralizes** the final hidden layer representation $h(\mathbf{x}) = \phi(\mathbf{x})$. Especially, in the **uniform $L_2$-feature penalty** case, we simply drop $\alpha$ in Eqn (12) and have $\mathbb{E}\{h\} = \mathbb{E}\{\phi(\mathbf{x})\} \to 0$.

In summary, the feature penalty **improves the numerical stability** of the optimization process in the regression problem. The conclusion also holds for the classification.

Similar results for $\phi(\mathbf{x})$ can be extended to deeper multilayer perceptrons with convex differentiable non-linear activation functions such as ReLU and max-pooling. In an $m$-layer model parametrized by $\{W_1, W_2, ..., W_m\}$, the Hessian matrix of hidden layers becomes strongly convex by back-propagating from the regularizer $||\sigma_{m-1}(W_{m-1} * \sigma_{m-2}(W_2 * (... * \sigma_1(W_1 \mathbf{x}))))||^2$.

### 2.3 UNIFORM $L_2$ FEATURE PENALTY IS A SOFT BATCH NORMALIZATION

In our two case studies, we can observe that $L_2$ feature penalty centers the representation $\phi(\mathbf{x})$, which reminds us of the *batch normalization* Ioffe & Szegedy (2015) with similar whitening effects. We reveal here that the two methods are indeed closely related in spirit.

From the Bayesian point view, we analyze a binary classification problem: the probability of prediction $\hat{y}$ given $\mathbf{x}$ observes a Bernoulli distribution $p(\hat{y}|\mathbf{x}, \phi, W) = \mathbf{Ber}(\hat{y}|\mathbf{sigm}(W\phi(\mathbf{x})))$, where $\phi(\mathbf{x}) \in \mathbb{R}^d$ is a neural network parametrized by $\phi$ and $W \in R^{1 \times d}$. Assuming a factorized Gaussian prior on $\phi(\mathbf{x}) \sim N(\mathbf{0}, \mathbf{Diag}(\sigma_1^2))$ and $W \sim N(\mathbf{0}, \mathbf{Diag}(\sigma_2^2))$, we have the posterior as:

$$p(\phi, W|\{(\mathbf{x}^i, y^i)\}) = p(\{(\mathbf{x}^i, y^i)\}|\phi, W)p(\phi)p(W)$$

$$\propto \prod_i (\frac{1}{1+e^{-W\phi(\mathbf{x}^i)}})^{y^i} (\frac{1}{1+e^{W\phi(\mathbf{x}^i)}})^{1-y^i} \frac{\exp(-\frac{||\phi(\mathbf{x}^i)||^2}{2\sigma_1^2})}{(\sqrt{2\pi}\sigma_1)^d} \frac{\exp(-\frac{||W||_F^2}{2\sigma_2^2})}{(\sqrt{2\pi}\sigma_2)^d} \tag{13}$$

Applying the maximum-a-posteriori (MAP) principle to Eqn(13) leads to the following objective:

$$\mathbb{E}\{l(W, \phi(\mathbf{x}^i), y^i) + \frac{1}{2\sigma_1^2}||\phi(\mathbf{x}^i)||^2\} + \frac{1}{2\sigma_2^2}||W||_F^2 + C \tag{14}$$

with $C = -d\ln(\sqrt{2\pi}\sigma_1) - d\ln(\sqrt{2\pi}\sigma_2)$ is a constant. This is exactly the uniform $L_2$ version of Equation (4) with $\lambda_1 = \frac{1}{2\sigma_1^2}$, $\lambda_2 = \frac{1}{2\sigma_2^2}$ and $\alpha^i = 1$. The *i.i.d.* Gaussian prior on $W$ and $\phi(\mathbf{x})$ has the effect of *whitening* the final representation.

The difference between uniform $L_2$ feature penalty and *batch normalization* is that the former whitens $\phi(\mathbf{x})$ implicitly with an *i.i.d.* Gaussian prior during training, while the latter explicitly normalizes the output by keeping moving average and variance. We could say that **the uniform $L_2$ penalty is a soft batch normalization**.

As analyzed in Wiesler & Ney (2011) this kind of whitening improved the numerical behavior and makes the optimization converge faster. In summary, the $L_2$ feature regularization on $\phi(x)$ indeed tends to reduce internal covariate shift.

## 2.4 FEATURE REGULARIZATION MAY IMPROVE GENERALIZATION

As pointed out in Hariharan & Girshick. (2016), the gradient penalty or feature regularization improves the performance of low-shot learning experimentally. Here, we give some preliminary analysis on how and why our modified model in Eqn 4 may improve generalization performance in neural network learning. Denoting the risk functional (testing error) $R(W)$ as:

$$R(W) = \frac{1}{2}\mathbb{E}\{||y - W\phi(x)||^2\} = \frac{1}{2}\int_{(\mathbf{x},y)\sim P(\mathbf{x},y)} ||y - W\phi(x)||^2 dP(\mathbf{x},y)$$

and empirical risk function (training error) $R_{emp}(W)$ on $\{(x^i, y^i)|i = 1, ..., N\}$ as:

$$R_{emp}(W) = \frac{1}{2N}\sum_{i=1}^{N} ||y^i - W\phi(x^i)||^2$$

As we know, the training and testing discrepancy depends on the model complexity Vapnik (1998):

$$R(W) - R_{emp}(W) \leq \nu(W) + O^*(\frac{h - \ln\eta}{N}) \tag{15}$$

where $O^*(.)$ denotes order of magnitude up to logarithmic factor. The upper bound 15 holds true with probability $1 - \eta$ for a chosen $0 < \eta < 1$. In the equation, $h$ is a non-negative integer named the VC-dimension. The right side of Eqn 15 contains two term: the first one $\nu(W)$ is the error on training set while the second one models complexity of the learning system. In a multilayer neural network with $\rho$ parameters and non-linear activations, the system has a VC dimension Sontag (1998) of $O(\rho\log\rho)$.

In our model in Eqn 4, the final cost function includes both feature and weight penalty as:

$$\arg\min_{W,\phi}\mathbb{E}\{l(W, \phi(\mathbf{x}^i), y^i) + \lambda_1\alpha^i||\phi(\mathbf{x}^i)||^2\} + \lambda_2||W||_F^2$$

Empirically, the term $\lambda_2||W||_F^2$ enforces large margin which limits the selection space of $W$. The term $\lambda_1||\phi(x^i)||^2$ not only improves numerical stability by feature whitening as discussed in the above sections, but also limits the selection of hidden layer parameter. These regularization terms thus reduces the VC-dimension of our learning. According to Eqn 15, the reduction of VC dimension of our model further reduces the theoretical discrepancy of training and testing performance.

## 3 EXPERIMENTAL RESULTS

We evaluate our algorithm on three datasets: synthetic XOR datasets, the Omniglot low-shot benchmark and the large-scale ImageNet dataset. We use our own implementation of SGM approach Hariharan & Girshick. (2016) for a fair comparison.

### 3.1 SYNTHETIC DATASETS

We first evaluate our model on the XOR dataset. Without loss of generality, we assume the data points $\mathbf{x} = [x_1, x_2]^T$ are uniformly sampled from a rectangle $x_1 \in [-1, 1], x_2 \in [-1, 1]$, and $y(\mathbf{x}) = \mathbb{I}(x_1 * x_2 > 0)$. The structure we use is a two-layer non-linear neural network with one

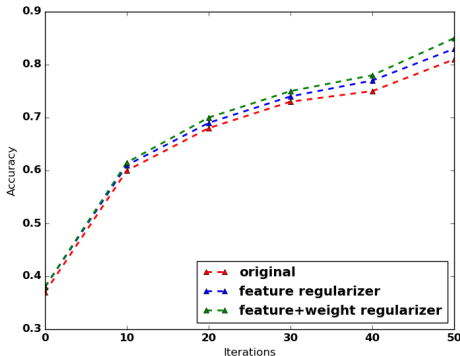

Figure 3: Evaluation on the XOR classification task. The red, blue and green lines stand for the accuracy of the Neural Network without any regularization, only the $L_2$ feature penalty, and our model with both weight and feature regularizer, respectively.

| Model | One-shot Evaluation |
|---|---|
| Random Guess | 5% |
| Pixel-KNN | 26.7% |
| MANN Santoro et al. (2016) | 36.4% |
| CNN Metric Learning | 72.9% |
| CNN (Our implementation) | 85.0% |
| Low-shot Hariharan & Girshick. (2016) | 89.5% |
| Matching Network Vinyals et al. (2016) | 93.8% |
| **Ours** | **91.5%** |

Table 1: Experimental results of our algorithm on the Omniglot benchmark Lake et al. (2011).

latent layer $h = \sigma(W_1 \mathbf{x} + \mathbf{b})$ where $\sigma(.)$ is the rectified linear unit. ADAM Kingma & Ba (2014) is leveraged to numerically optimize the cross entropy loss.

During training, we make use of only 4 points $\{[1,1], [-1,1], [1,-1], [-1,-1]\}$, while randomly sampled points from the whole rectangle as the test set. It is a very low-shot task. As shown in Figure 3, our model with both feature and weight regularization outperforms the gradient penalty Hariharan & Girshick. (2016) and no regularization. We use uniform feature penalty as in Eqn(4) and set $\lambda_1 = \lambda_2 = 0.1$ in our experiments.

## 3.2 LOW-SHOT LEARNING ON OMNIGLOT

Our second experiment is carried out on the Omniglot one-shot benchmark Lake et al. (2011). Omniglot training set contains 964 characters from different alphabets with only 20 examples per each character. The one-shot evaluation is a pairwise matching task on completely unseen alphabets.

Following Vinyals et al. (2016), we use a simple yet powerful CNN as feature representation model, consisting of a stack of modules, each of which is a $3 \times 3$ convolution with $128$ filters followed by batch normalizationIoffe & Szegedy (2015), a ReLU and $2 \times 2$ max-pooling. We resized all images to $28 \times 28$ so that the resulting feature shape satisfies $\phi(x) \in \mathbb{R}^{128}$. A fully connected layer followed by a softmax non-linearity is used to define the Baseline Classifier.

We set $\lambda_1$=1e-4 in SGM Hariharan & Girshick. (2016) and $\lambda_1$=$\lambda_2$=1e-4 in our model. A nearest neighbor approach with $L_2$ distance of feature $\phi(\mathbf{x})$ is applied for one-shot evaluation. As shown in Table 1, we can see that our model with both feature and weight penalty is able to achieve satisfactory performance of one-shot $91.5\%$ accuracy, highly competitive with the state-of-the-art Matching NetworkVinyals et al. (2016) with CNN warm-start and RNN hyper-parameter tuning.

| Model | One-shot Evaluation |
|---|---|
| CNN baseline | 40.1% |
| Low shot Hariharan & Girshick. (2016) | 46.0% |
| **Ours** | **46.6%** |

Table 2: Experimental results of our algorithm on the ImageNet benchmark Russakovsky et al. (2015) with the 20-way one-shot setting.

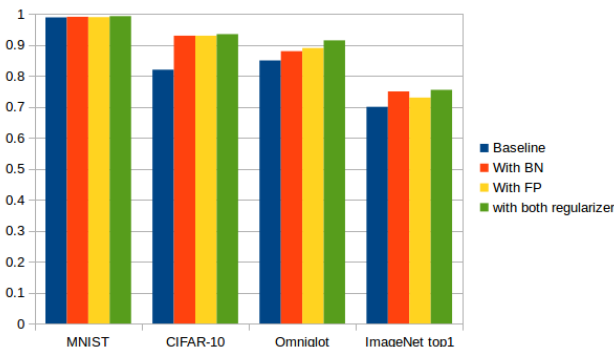

Figure 4: Classification accuracy comparison of our feature penalty (termed "PN") with batch-normalization (termed "BN") Ioffe & Szegedy (2015) on MNIST, CIFAR-10, Omniglot and ImageNet benchmarks. We compare baseline methods with neither batch-normalization nor feature penalty, with each module added, and with modules included.

### 3.3 LARGE-SCALE LOW-SHOT LEARNING ON IMAGENET

Our last experiment is on the ImageNet benchmark Russakovsky et al. (2015). It contains a wide array of classes with significant intra-class variation. We divide the 1000 categories randomly into 400 base for training and evaluate our feature representation on the 600 novel categories.

We use a 50-layer residual network He et al. (2016) as our baseline. Evaluation is measured by top-1 accuracy on the 600 test-set in a 20-way setup, i.e., we randomly choose 1 sample from 20 test classes, and applies a nearest neighbor matching. As shown in Table 2, we can see that our model learns meaningful representations for unseen novel categories even with large intra-class variance.

### 3.4 COMPARISON WITH BATCH-NORMALIZATION

As discussed in Section 2.3, feature penalty has similar effects with batch normalization Ioffe & Szegedy (2015). It is of interest to compare the influence of two modules influence training performance of neural networks. We study the performance of the classification with and without each modules on four classic image classification benchmarks. For CIFAR-10 and ImageNet, we applied the Residual Net architecture He et al. (2016), while stacked convolution layers with ReLU and max-pooling is applied for MNIST and Omniglot. For ImageNet benchmark evaluation, we test the top-1 accuracy on the validation set with 50,000 images.

Since in our model the feature penalty regularizer is applied only on the last hidden layer, we still keep the batch-normalization modules in previous layers in our "FP" model. As shown in Figure 4, we observe that baseline models with neither "BN" nor "FP" takes much longer to converge and achieve inferior performance; our "FP" regularizer achieves almost the same performance on MNIST (both 99%), CIFAR-10 (both 93%) and Omniglot 1-shot (88% BN v.s. 89% FP); on ImageNet, "BN" performs better than our "FP" (75% BN v.s. 74% FP). With both batch-normalization

and feature penalty modules added, we achieve the best classification performance on all four benchmarks.

## 4 CONCLUSION

In this work, we conduct an analysis, both empirically and theoretically, on how feature regularization influences and improves low-shot learning performance. By exploiting an XOR classification and two-layer linear regression, we find that the regularization of feature centers the representation, which in turn makes the learning problem easier with better numerical behavior. From the Bayesian point of view, the feature regularization is closely related to batch normalization. Evaluation on synthetic, "Omniglot" one-shot and large-scale ImageNet benchmark validates our analysis.

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
