# Peer review of "An Analysis of Feature Regularization for Low-shot Learning"

_ICLR 2017 — rejected_

[Official Review · AnonReviewer2 · rating 5 · confidence 4 · 15 Dec 2016 (modified: 12 Jan 2017)]

The paper proposes to use a last-layer feature penalty as regularization on the last layer of a neural net.
Although the equations suggest a weighting per example, dropping this weight (alpha_i) works equally well.
The proposed approach relates to Batch Norm and weight decay.
Experiments are given on "low-shot" settting.
There seem to be two stories in the paper: feature penalty as a soft batch norm version, and low-shot learning; why is feature penalty specifically adapted to low-shot learning and not a more classical supervised task?
Regarding your result on Omniglot, 91.5, I believe it is still about 2% worse than the Matching Networks, which you refer to but don't put in Table 1. Why?
Overall, the idea is simple but feels like preliminary: while it is supposed to be a "soft BN", BN itself gets better performance than feature penalty, and both together give even better results. Is something still missing in the explanation?

-- edits after revised version:

Thank you for adding more information to the paper. I feel it is still too long but hopefully you can reduce it to 9 pages as promised. However, I'm still not convinced the paper is ready to be accepted, mainly for the following reasons:
- on Omniglot, the paper is still significantly far from the current state of the art.
- the new experiments do not really confirm/infirm the relationship with BN.
- you added an explanation of why FP works for low-shot setting, by showing it controls the VC dimension and hence is good to control overfitting with a small number of training examples, but this discussion is basic and does not really shed more light than the obvious.
I'm pushing up your score from 4 to 5 for the improved version, but I still think it is below acceptance level.

[Official Review · AnonReviewer1 · rating 6 · confidence 3 · 16 Dec 2016]
**No Title**

This paper proposes analysis of regularization, weight Froebius-norm and feature L2 norm, showing that it is equivalent to another proposed regularization, gradient magnitude loss. They then argue that: 1) it is helpful to low-shot learning, 2) it is numerically stable, 3) it is a soft version of Batch Normalization. Finally, they demonstrate experimentally that such a regularization improves performance on low-shot tasks.

First, this is a nice analysis of some simple models, and proposes interesting insights in some optimization issues. Unfortunately, the authors do not demonstrate, nor argue in a convincing manner, that such an analysis extends to deep non-linear computation structures. I feel like the authors could write a full paper about "results can be derived for φ(x) with convex differentiable non-linear activation functions such as ReLU", both via analysis and experimentation to measure numerical stability.

Second, the authors again show an interesting correspondance to batch normalization, but IMO fail to experimentally show its relevance.

Finally, I understand the appeal of the proposed method from a numerical stability point of view, but am not convinced that it has any effect on low-shot learning in the high dimensional spaces that deep networks are used for.

I commend the authors for contributing to the mathematical understanding of our field, but I think they have yet to demonstrate the large scale effectiveness of what they propose. At the same time, I feel like this paper does not have a clear and strong message. It makes various (interesting) claims about a number of things, but they seem more or less disparate, and only loosely related to low-shot learning.

notes:
- "an expectation taken with respect to the empirical distribution generated by the training set", generally the training set is viewed as a "montecarlo" sample of the underlying, unknown data distribution \mathcal{D}.
- "we can see that our model learns meaningful representations", it gets a 6.5% improvement on the baseline, but there is no analysis of the meaningfulness of the representations.
- "Table 13.2" should be "Table 2".
- please be mindful of formatting, some citations should be parenthesized and there are numerous extraneous and missing spacings between words and sentences.

[Official Review · AnonReviewer3 · rating 6 · confidence 3 · 20 Dec 2016]
**Interesting analysis, though a clear theoretical relationship between feature regularization and low-shot learning seems missing**

Summary
===
This paper extends and analyzes the gradient regularizer of Hariharan and
Girshick 2016. In that paper a regularizer was proposed which penalizes
gradient magnitudes and it was shown to aid low-shot learning performance.
This work shows that the previous regularizer is equivalent to a direct penalty
on the magnitude of feature values weighted differently per example.

The analysis goes to to provide two examples where a feature penalty
favors a better representation. The first example addresses the XOR
problem, constructing a network where a feature penalty encourages
a representation where XOR is linearly separable.
The second example analyzes a 2 layer linear network, showing improved stability
of a 2nd order optimizer when the feature penalty is added.
One last bit of analysis shows how this regularizer can be interpreted as
a Gaussian prior on both features and weights. Since the prior can be
interpreted as having a soft whitening effect, the feature regularizer
is like a soft version of Batch Normalization.

Experiments show small improvements on a synthetic XOR test set.
On the Omniglot dataset feature regularization is better than most baselines,
but is worse than Moment Matching Networks. An experiment on ImageNet similar
to Hariharan and Girshick 2016 also shows effective low-shot learning.


Strengths
===

* The core proposal is a simple modification of Hariharan and Girshick 2016.

* The idea of feature regularization is analyzed from multiple angles
both theoretically and empirically.

* The connection with Batch Normalization could have broader impact.


Weaknesses
===

* In section 2 the gradient regularizer of Hariharan and Girshick is introduced.
While introducing the concept, some concern is expressed about the motivation:
"And it is not very clear why small gradients on every sample produces
good generalization experimentally." This seems to be the central issue to me.
The paper details some related analysis, it does not offer a clear answer to
this problem.


* The purpose and generality of section 2.1 is not clear.

The analysis provides a specific case (XOR with a non-standard architecture)
where feature regularization intuitively helps learn a better representation.
However, the intended take-away is not clear.

The take-away may be that since a feature penalty helps in this case it
should help in other cases. I am hesitant to buy that argument because of the
specific architecture used in this section. The result seems to rely on the
choice of an x^2 non-linearity, which is not often encountered in recent neural
net literature.

The point might also be to highlight the difference between a weight
penalty and a feature penalty because the two seem to encourage
different values of b in this case. However, there is no comparison to
a weight penalty on b in section 2.1.


* As far as I can tell, eq. 3 depends on either assuming an L2 or cross-entropy
loss. A more general class of losses for which eq. 3 holds is not provided. This
should be made clear before eq. 3 is presented.


* The Omniglot and ImageNet experiments are performed with Batch Normalization,
yet the paper points out that feature regularization may be similar in effect
to Batch Norm. Since the ResNet CNN baseline includes Batch Norm and there are
clear improvements over that baseline, the proposed regularizer has a clear
additional positive effect. However, results should be provided without
Batch Norm so a 1-1 comparison between the two methods can be performed.


* The ImageNet experiment should be more like Hariharan and Girshick.
In particular, the same split of classes should be used (provided in
the appendix) and performance should be measured using n > 1 novel examples
per class (using k nearest neighbors).


Minor:

* A brief comparison to Matching Networks is provided in section 3.2, but the
performance of Matching Networks should also be reported in Table 1.

* From the approach section: "Intuitively when close to convergence, about half
of the data-cases recommend to update a parameter to go left, while
the other half recommend to go right."

Could the intuition be clarified? There are many directions in high
dimensional space and many ways to divide them into two groups.

* Is the SGM penalty of Hariharan and Girshick implemented for this paper
or using their code? Either is acceptable, but clarification would be appreciated.

* Should the first equal sign in eq. 13 be proportional to, not equal to?

* The work is dense in nature, but I think the presentation could be improved.
In particular, more detailed derivations could be provided in an appendix
and some details could be removed from the main version in order to increase
focus on the results (e.g., the derviation in section 2.2.1).


Overall Evaluation
===

This paper provides an interesting set of analyses, but their value is not clear.
There is no clear reason why a gradient or feature regularizer should improve
low-shot learning performance. Despite that, experiments support that conclusion,
the analysis is interesting by itself, and the analysis may help lead to a
clearer explanation.

The work is a somewhat novel extension and analysis of Hariharan and Girshick 2016.
Some points are not completely clear, as mentioned above.

[Author Response · Zhuoyuan Chen · 11 Jan 2017]
**A new revised version just updated on Jan 11th**

We would like to thank all reviewers for their valuable comments and suggestions. We modify our paper accordingly. 
Especially, we add classification performance comparison between our methods and batch normalization.
Moreover, we also add our preliminary analysis why the proposed regularizer could improve generalization performance, which is closely related to the improved "low-shot" improvement.

Admittedly, current version is a little bit beyond page-limit (11 pages now). We managed to include some new experimental results and analysis in our revised paper, and will trim it within 9 pages with some detailed deductions left in supplemental materials.

[Final Decision · Program Chairs · 06 Feb 2017]
**ICLR committee final decision**

The paper extends a regularizer on the gradients recently proposed by Hariharan and Girshick. I agree with the reviewers that while the analysis is interesting, it is unclear why this particular regularizer is especially relevant for low-shot learning. And the experimental validation is not strong enough to warrant acceptance.